# Survival with Good Neurological Outcome despite Prolonged Cardiopulmonary Resuscitation and Extreme Acidosis after Out-of-Hospital Cardiac Arrest Due to Acute Myocardial Infarction: A Case Report and Review of the Literature

Sylvère Störmann [1,*], Kristina Busygina [1], Ralph Hein-Rothweiler [2], Julius Steffen [2], Stefanie Förderreuther [3], Nora Salein [4] and Matthias W. Angstwurm [1]

1 Medizinische Klinik und Poliklinik IV, Klinikum der Universität München, LMU München, 80336 Munich, Germany
2 Medizinische Klinik und Poliklinik I, Klinikum der Universität München, LMU München, 81377 Munich, Germany
3 Neurologische Klinik und Poliklinik, Klinikum der Universität München, LMU München, 81377 Munich, Germany
4 Klinik für Anästhesiologie und Intensivmedizin, München Klinik, Klinikum München Schwabing, 80804 Munich, Germany
* Correspondence: sylvere.stoermann@med.uni-muenchen.de; Tel.: +49-89-4400-52318

**Abstract:** We report the case of a 49-year-old male who suffered from a myocardial infarction with subsequent cardiac arrest. The emergency medical team began cardiopulmonary resuscitation, including defibrillation of ventricular fibrillation. Although a return of spontaneous circulation was achieved after approximately 30 min of continued efforts, the patient went back into cardiac arrest on the way to the hospital and resuscitation had to be resumed. On admission, the patient was severely acidotic with a pH of 6.67, lactatemia of 19 mmol/L, and pronounced hypercapnia ($pCO_2$ 127 mmHg). Despite the poor prognosis, all possible efforts including coronary intervention and therapeutic hypothermia were carried out and the patient made a quick recovery with discharge from the intensive care unit on day 5. Survival of extreme acidosis, such as in this case, is rare. This is the first report of survival with good neurologic outcome in a patient with myocardial infarction, cardiac arrest, and pH of under 6.7 on admission at the clinic.

**Keywords:** cardiac failure; acidosis; resuscitation; myocardial infarction



## 1. Introduction

Coronary heart disease is the leading cause of death in the developed world. Its most common form of presentation, myocardial infarction (MI), has an incidence of more than 7 million cases worldwide every year [1]. Despite advances in diagnosis and therapy, it still is considered highly lethal, in particular when it is complicated with cardiac arrest. Patients with cardiac arrest in out-of-hospital settings survive in only 10.6% of cases, a quota that is greatly improved to 33.6% in bystander-witnessed arrests with shockable rhythms, though nonetheless remaining highly lethal [2]. Cardiac arrest causes ischemia of tissue and cells and thus leads to a metabolic imbalance of acid–base homeostasis. Furthermore, it inevitably leads to respiratory failure, which further worsens acidosis. The acid–base homeostasis is a finely tuned system of the body's extracellular fluid and a chemical requirement for the correct functioning of protein folding, which in turn affects practically all bodily functions. An acid–base imbalance therefore leads to changes in receptor configurations, ion distribution, and enzyme performance. To this end, several mechanisms in humans tightly regulate pH within very narrow limits around a value of 7.4. Outside this scope, most of the medications used in emergencies, such as norepinephrine and epinephrine, have attenuated efficacy, making extreme deviations practically untreatable.

Here, we present the case of a patient who survived nominally lethal pH levels after cardiac arrest due to myocardial infarction.

## 2. Case Report

A 49-year-old employee of Munich municipal services felt a sudden onset of severe chest pain and asked the gatekeeper of a nearby public parking garage for medical help. The patient had previously led a medically unremarkable life without any known health issues and no relevant family history. He did not take any medication. However, he did consume alcohol regularly (approximately 3–4 beers a week) and was a heavy and long-time cigarette smoker (30 cigarettes per day with about 30 to 40 pack years). He had never consumed any other drugs. He spent most of his days outdoors and walked a lot. He did not overeat and was of normal weight. Shortly after seeking help, he collapsed and remained unresponsive. No bystander cardiopulmonary resuscitation (CPR) was performed. When the emergency medical services arrived 16 min after the initial emergency call, the patient was unconscious with no reaction to pain (Glasgow Coma Scale score 3), had widened pupils not reactive to light, and was apneic. The initial heart rhythm was ventricular fibrillation. The emergency physician performed eight consecutive defibrillations and applied a total of 9 mg epinephrine, 600 µg norepinephrine, 600 mg amiodarone, 20 mmol magnesium, and 100 mg lidocaine due to persistent ventricular fibrillation. A mechanical chest compression system (LUCAS®, Stryker GmbH & Co. KG, Duisburg, Germany) was installed. Airway management proved difficult and endotracheal intubation failed at several attempts despite the long-standing experience of the emergency physician. Finally, a laryngeal tube was inserted. Oxygenation was sufficient and end-tidal $CO_2$ values were around 40 mmHg, suggesting sufficient resuscitation efforts. After a total of 38 min of CPR, a ventricular escape rhythm set in and spontaneous circulation returned with palpable peripheral pulse. Only 5 min later, ventricular fibrillation recurred and could be treated with one more defibrillation (300 J) resulting in sinus bradycardia. The emergency team prepared for transportation to the nearest hospital. Just after arriving in the emergency vehicle, the patient's pulse and circulation stopped again due to ventricular fibrillation, and CPR had to resume, including defibrillation while driving.

On arrival at our intensive care unit 68 min after the initial emergency call, the patient still had dilated and unresponsive pupils, pulseless electrical activity, and bradycardia (<30/min). The ECG recording did not reveal ST elevations. Arterial blood gas analysis revealed severe combined metabolic and respiratory acidosis (pH 6.67; $pCO_2$ 127 mmHg; lactate 19 mmol/L; base excess −20.5 mmol/L; $etCO_2$ was recorded as 42 mmHg) with mildly subnormal core temperature (35.3 °C). Due to hypercapnia, sodium bicarbonate was not applied and buffering with trometamol was briefly considered. Instead, an endotracheal intubation was undertaken and was successful. After another 5 min of CPR and continuous administration of epinephrine and vasopressin, ROSC was finally achieved. $pCO_2$ quickly dropped to 80.7 mmHg. The patient was then immediately transferred to the catheterization laboratory.

Coronary angiography revealed signs of a plaque rupture in the proximal left anterior descending artery, which was successfully treated with an angioplasty stenting procedure implanting a drug-eluting stent. During the procedure, adequate ventilation was maintained and $pCO_2$ continually decreased while high doses of epinephrine and norepinephrine (1 mg/h each) maintained circulation with decreasing lactate levels. Consequently, pH improved. However, the implantation of an extracorporeal life support device (ECLS) was not considered due to a presumably poor prognosis. On echocardiography, a severely impaired left ventricular systolic function with hypokinesia of the anterior, apical, and septal walls was found. Enoximone-induced phosphatidysterase inhibition was administered, leading to a significant improvement in left ventricular function. Upon return from the catheterization laboratory, the pH had risen to 7.11 in the arterial blood ($pCO_2$ 50.1 mmHg; lactate 9.5 mmol/L; base excess −12.5 mmol/L). We applied targeted temperature management and the patient's minimal temperature of 33.6 °C was reached approximately 4 h after admission. Hypothermia in the range of 33 to 36 °C was maintained

over about 12 h, and subsequently the temperature target was kept below 37 °C. After the catheter procedure, sodium bicarbonate was finally applied twice (8.4 g each) at 5 and 11 h after admission. Ultimately, acid–base balance was restored within 12 h after admission (arterial blood; pH 7.37; $pCO_2$ 35.7 mmHg; lactate 5.1 mmol/L; base excess −3.6 mmol/L). Initial troponine measurement was 0.264 ng/mL (reference range: <0.013 ng/mL) and peaked at 1.87 ng/mL. Peak creatine kinase was 15,566 U/L (normal range: ≤189 U/L).

During the first 48 h after infarction, the patient developed severe reperfusion arrhythmias. Intravenous amiodarone treatment was continued and beta blockade was started. Electrolytes were kept in the normal range. The ECG rhythm stabilized and we discontinued amiodarone treatment on day 4.

Lab work revealed signs of multi-organ failure with elevated creatinine and transaminase levels, as well as impaired hemostasis as signs of cardiogenic shock. However, organ function quickly recovered. Protein S100 levels were elevated at admission (3.8 ng/mL) and neuron-specific enolase rose to a maximum of 47.6 ng/mL on d2; both parameters decreased over the next few days. The patient was quickly weaned from the ventilator and extubation was possible on day 3. That same day, the patient uttered scattered words and seemed to have the urge to convey something with hardly recognizable words. However, the neurologic status improved rapidly: the patient started saying complete sentences and was able to walk a few steps on day 4. During rounds on day 5 the patient pointed a finger at his chest and asked: "This is my motor, right?" We replied: "You might say so." He grinned and joyfully proclaimed: "Then, it seems as if I just passed my vehicle inspection!" We concluded that the patient no longer needed our intensive care and discharged him to the cardiology ward.

In a follow-up echocardiography on day 5, left ventricular function was estimated at around 45%. Consequently, cardiac magnetic resonance imaging showed cardiomegaly, apical hypokinesis, and ballooning of the left ventricle. The patient recovered further and was discharged from the hospital on day 11 to undergo rehabilitation.

## 3. Discussion

Right from the start, the survival of our patient was very unlikely. Our patient was 49 years old when he suffered his first myocardial infarction. In his age group, coronary heart disease occurs in 3.4% of men [3]. The German Federal Statistical Office provides detailed data on the incidence and mortality of acute myocardial infarction collected in the MONICA/KORA registry. According to this data, between 2015 and 2017, the annual rate of myocardial infarction was 185 in 100,000 males aged 45 to 49. Of those, 21.5% were lethal. Recently, a "CaRdiac Arrest Survival Score (CRASS)" was proposed using multiple predictors such as age, rhythm, duration of CPR, and other factors from the prehospital setting to predict hospital outcome after out-of-hospital cardiac arrest [4]. According to this tool, our patient had a predicted survival with a good neurological outcome of 6.3%.

Early work estimating survival from empirical data published in the 1970s was based on lactate levels [5] and predicts a similarly poor outcome given the high lactate of the patients of 19 mmol/L. In data from clinical experience, lactate is a strong predictor of survival [6]. In addition to that, a profound hypercapnia was found upon arrival on our ICU. End-tidal $CO_2$ levels did not accurately reflect the systemic partial pressure of carbon dioxide due to insufficient cardiac output. Furthermore, airway management using a laryngeal tube is associated with a worse outcome, as could be shown in an analysis of the German Resuscitation Registry [7].

Admittedly, the dismal prognosis left us little hope of achieving a positive outcome. In patients presenting with a pH < 6.7, survival is a rarity, let alone a good neurological outcome. In a retrospective multicenter analysis of 2229 outpatient cardiac arrest patients, Shin and colleagues found a survival rate to hospital discharge of 14% [8]. The initial pH level at arrival in the emergency department was an independent predictor of survival (OR 6.3 [95% CI: 2.6–15.2]). Only 1 in 31 patients (3.2%) survived a pH below 6.7, but none of the patients with pH < 6.8 had a good neurological outcome. Similarly, Chien et al. found

a positive correlation bewteen initial pH and survival [9]. Interestingly, the best receiver operating characteristic curve was obtained with a cut-off of 7.07; none of the patients with a pH below 6.85 survived. Momiyama and colleagues obtained similar results and found a pH of 7.05 to be the optimal cut-off to predict a good neurological outcome, which none of the patients with pH < 6.95 achieved [10]. In another study of thirty patients with cardiac arrest and pH < 7.0, only three survived [11]. Indeed, the cardiac arrest in these survivors occurred in the presence of a medical team so that resuscitation began immediately without delay. However, in a smaller study comprising 25 patients, Paz and colleagues showed that extreme pH values do not necessarily mean certain death, with survivors with pH as low as 6.6 [12]. It must be noted that none of the patients with multiorgan failure or after cardiopulmonary resuscitation survived.

This is an important issue: Does the etiology of acidosis play a role in the survival of extreme pH levels [13]? Only few cases of survival of pH levels below 6.7 exist in the literature. Most published case reports relate to intoxication events, including instances of strychnine and metformin-associated lactic acidosis. Other prevalent scenarios involve acute hemorrhage with circulatory shock, and episodes of diabetic ketoacidosis. None of these involve cardiac arrest in myocardial infarction. Likewise, a study from Ireland identified 130 patients with pH < 7.0 in a 4-year period in the emergency department of the University Hospital of Galway and found a high mortality rate, particularly in patients with cardiac arrest [14]. This indicates possibly favorable outcomes even in severe acidosis due to intoxication and other cases of metabolic acidosis not limited to poisoning. In some of the cases in the literature, hypothermia was extreme and may have played an important protective factor. In our case, resuscitation took place primarily in a parking garage, i.e., in a cold environment, and the patient presented with mild hypothermia. After admission, we applied targeted temperature management. Hypothermia reduces metabolic rate and oxygen consumption, allowing the brain to better tolerate ischemic insults caused by cardiac arrest, thus reducing the extent of neurological damage and thus improving neurological outcomes. Furthermore, there is evidence of neuroprotective effects of hypercapnia, since carbon dioxide decreases cerebrovascular resistance and thus increases blood flow [15]. However, it should be noted that $PaCO_2$ has a U-shaped inverted association with a good neurological outcome, which worsens in the presence of metabolic acidosis. Considering the data from our case, the prognosis would still have to be judged as dismal.

Of note, it has been shown that a history of smoking is associated with improved neurological outcome in patients undergoing therapeutic hypothermia after cardiac arrest [16]. Our patient had an excessive smoking history of approximately 50 pack years.

To our knowledge, this is the first report of a patient with myocardial infarction with cardiac arrest and a pH < 6.7 and therefore unfavorable prognosis who not only survived but recovered quickly and without any apparent sequelae. A similar case was published with out-of-hospital cardiac arrest, prolonged CPR, low pH, and high lactate levels who stabilized after percutaneous coronary intervention and surprised the medical team with quick recovery and a good neurological outcome [17]. This shows that patients with myocardial infarction may have a good outcome despite a desolate initial assessment according to established criteria and scoring systems.

### 4. Conclusions

Survival of extreme acidosis per se is rare, and even less common in cases due to cardiac arrest after myocardial infarction. Our report demonstrates that—even in light of a dim prognosis—survival with a good neurologic outcome in a patient with myocardial infarction, cardiac arrest, and pH of under 6.7 on admission at the clinic is possible.

**Author Contributions:** Conceptualization, S.S., J.S. and M.W.A.; formal analysis, R.H.-R. and S.F.; investigation, S.S., K.B. and M.W.A.; resources, N.S.; writing—original draft preparation, S.S.; writing—review and editing, J.S. and M.W.A.; supervision, M.W.A. All authors have read and agreed to the published version of the manuscript.

**Funding:** This research received no external funding.

**Institutional Review Board Statement:** Our institution's Ethics Committee does not require ethical review and approval of case reports.

**Informed Consent Statement:** Written informed consent has been obtained from the patient to publish this report.

**Data Availability Statement:** The data pertaining to this case are presented within the article. Other data are not publicly available due to regulations.

**Conflicts of Interest:** The authors declare no conflict of interest.

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
