# Peer review of "Survival with Good Neurological Outcome despite Prolonged Cardiopulmonary Resuscitation and Extreme Acidosis after Out-of-Hospital Cardiac Arrest Due to Acute Myocardial Infarction: A Case Report and Review of the Literature"

_clinpract, doi:10.3390/clinpract13030056_

Round 1

Reviewer 1 Report

1. Despite the very serious condition of this case after cardiac arrest resuscitation, the patient regained consciousness. However, there was nothing noteworthy about the treatment. It is not at all clear what led to the favorable outcome. I cannot find any academic value in this as a paper. I would like to hear the author's opinion.

 2. It appears that TTM was not performed on this patient. Does the author's institution not perform TTM after cardiac arrest resuscitation?

 3. The cases listed in Table 1 are not cardiac arrest cases. Even if the acidosis is severe, the outcome is different between metabolic disease and cardiac arrest. The table is unnecessary and should be deleted.

Reviewer 2 Report

This is a very interesting case concerning out-of-hospital cardiac arrest with critical neurological status and spectacular recovery. The paper is logical, well-written with a nice review of the literature. However, there are certain issues that should be clarified.

1. The clinical profile of the patient is quite undiscovered in the paper. We know his age, manual worker and a heavy smoker. Did he have any chronic diseases or positive family history, did he use any medication, psychotropic medicines, alcohol, stimulants etc.?

2. Was not sodium bicarbonate used by the emergency physician on at the intensive care unit?!

3. How was body temperature controlled? In the abstract section one can read that therapeutic hypothermia was carried out, while in the case report section we are provided with information that initial hypothermia was not actively induced but temperature (body temperature) was kept below 37 degrees Celsius. This should be discussed, why the patient did not receive therapeutic hypothermia with targeted 32 – 34 degrees Celsius and what implications therapeutic hypothermia has on neurological recovery after sudden cardiac arrest.

4. What sort of blood was used for measurements - arterial, arterialized venous, venous and capillary blood in your case and in the overviewed published reports?

5. Did you check blood alcohol concentration?

6. Which age is correct – 47 from the abstract or 49 from the case presentation?

7. What was recorded in ECG?

8. Why the patient was not transferred directly from the ambulance to the hemodynamics laboratory?

9. I do not think that a plaque rupture can be seen on simple coronary angiography.  

10. I think that the correct name of the procedure is “angioplasty stenting procedure” (with drug eluting stent implantation?). Balloon angioplasty is a procedure finished with no stent.

11. Was it just hypokinesis in MRI? Was the term ‘ballooning of the left ventricle’ used in the MRI description, which shows hypercontraction of the basal segment (met in Tako-tsubo cardiomyopathy?)

12. What was the blood pressure of the patient?

13. What were other blood results (troponin, creatinine, ions or for example methemoglobin, CRP)?

14. Was there any pharmacotherapy used for neurological recovery? What in the pharmacological treatment of myocardial infarction could affect the neurological recovery? I will read about this with interest.

15. What is the practical benefit of monitoring of Protein S100 or neuron-specific enolase?

16. My impression after reading the manuscript concern cardiopulmonary vs pure chest compressions, availability of AEDs, resuscitation tutorial at school and work places, use of all available means it hospital and health status monitoring of middle-aged people.

Reviewer 3 Report

The authors presented a case of survival with extreme acidosis (pH 6.67) after prolonged cardiopulmonary resuscitation due to ischemic cardiac arrest in a 49-year old male. The patient was complicated with multi-organ failure but despite that almost fully recovered.  This is and added case in the literature of good outcomes in such extreme conditions.  Takin this case as an example the resuscitators/operators should be encouraged to perform a  good CPR and not to lose optimism.

Minor Issues

Please improve in line 29 “7 million times” with for example “cases” instead.

The incidence of new MI 7 million/year worldwide and please add this information. If you find some more recent statistic (that is from 2008) you can update.

Line 31 – “ in particular when it comes to cardiac arrest” > “ in particular when it is complicated with cardiac arrest”

Round 2

Reviewer 1 Report

I have peer reviewed the revised paper. Unfortunately, the authors have not complied with my request.

1. Even if a post-cardiac arrest resuscitation case has advanced metabolic acidosis on arrival, acute treatment should not be given up. Intensive care, including ventilation, circulatory management, and TTM, is commonly provided for postresuscitation cases. In addition, coronary intervention is the obvious treatment for postresuscitation cases caused by acute myocardial infarction. In the present case, no special therapeutic intervention was performed. From these points of view, there is no novelty in this paper.

2. As pointed out previously, TABLE 1 is irrelevant to this paper. If you want to describe a case of severe acidosis other than cardiac arrest, you should write another paper.

Author Response

Dear Editors and Reviewers,

I hope this message finds you well.

Firstly, I would like to express my profound gratitude for your time and effort spent reviewing and rereading our manuscript. Your insightful comments and suggestions have greatly contributed to the enhancement of the overall quality of our work. We are particularly thankful for your patience and diligence in this process.

Upon reviewing your comments, we noticed the oversight regarding Table 1, which we previously mentioned in our rebuttal letter would be removed. We sincerely apologize for any confusion this might have caused. We found this table useful to illustrate how rare survival with such profound acidosis is and that this usually occurs in another clinical context (mostly intoxication). As you have rightly pointed out, the manuscript would indeed be clearer and more succinct without it.

Regrettably, we cannot quite explain how this oversight occurred, and we are sincerely sorry for any inconvenience this might have caused. We have now properly removed Table 1 from the manuscript and made corresponding modifications to the body text to ensure clarity and coherence of the content.

In regard to the concern raised about the novelty of our study, we appreciate your perspective and understand the importance of novelty in advancing the scientific discourse. However, we believe the novelty of our study lies in the unique clinical situation and its outcome. Indeed, our manuscript represents the first report of survival with a favorable neurologic outcome in a patient presenting with myocardial infarction, cardiac arrest, and an extremely low pH of under 6.7 on admission to our clinic. To the best of our knowledge, no such outcome has been previously documented in a cardiac arrest patient presenting with such extreme acidosis. Therefore, we believe our work provides good documentation of an unusual clinical situation and course, even if arguably the management was based on standard care. We hope that this clarification adequately addresses your concerns regarding the novelty of our report.

Once again, we extend our sincere thanks for your valuable comments and the time you have dedicated to our manuscript. We are confident that these revisions have significantly improved our paper and we look forward to any additional comments you may have.

Regards,

Sylvère Störmann on behalf of the authors.